# A Bayesian Approach for Modeling and Forecasting Solar Photovoltaic Power Generation

**DOI:** 10.3390/e26100824

**Published:** 2024-09-27

**Authors:** Mariana Villela Flesch, Carlos Alberto de Bragança Pereira, Erlandson Ferreira Saraiva

**Affiliations:** 1Faculty of Engineering, Architecture and Urbanism and Geography, Federal University of Mato Grosso do Sul, Campo Grande 79070-900, MS, Brazil; mariflesch@hotmail.com; 2Institute of Matematics and Statistics, University of São Paulo, São Paulo 05508-090, SP, Brazil; cadebp@gmail.com; 3Institute of Matematics, Federal University of Mato Grosso do Sul, Campo Grande 79070-900, MS, Brazil

**Keywords:** photovoltaic solar power forecasting, statistical modeling, Bayesian inference, Gaussian process, MCMC, Gibbs sampling algorithm

## Abstract

In this paper, we propose a Bayesian approach to estimate the curve of a function f(·) that models the solar power generated at *k* moments per day for *n* days and to forecast the curve for the (n+1)th day by using the history of recorded values. We assume that f(·) is an unknown function and adopt a Bayesian model with a Gaussian-process prior on the vector of values $f\left(t\right)=\left(f(1),\ldots ,\;f(k)\right)$. An advantage of this approach is that we may estimate the curves of f(·) and $f_{n+1}\left(\cdot \right)$ as “smooth functions” obtained by interpolating between the points generated from a *k*-variate normal distribution with appropriate mean vector and covariance matrix. Since the joint posterior distribution for the parameters of interest does not have a known mathematical form, we describe how to implement a Gibbs sampling algorithm to obtain estimates for the parameters. The good performance of the proposed approach is illustrated using two simulation studies and an application to a real dataset. As performance measures, we calculate the absolute percentage error, the mean absolute percentage error (MAPE), and the root-mean-square error (RMSE). In all simulated cases and in the application to real-world data, the MAPE and RMSE values were all near 0, indicating the very good performance of the proposed approach.

## 1. Introduction

In recent years, there has been a significant increase in solar energy generation, from both photovoltaic plants and residences with photovoltaic panels installed on their roofs. This has been driven by societal and governmental interest in clean and renewable energy, with an important aspect of “clean” being lower ${\mathrm{CO}}_{2}$ emission compared to fossil fuels. Because of this, more photovoltaic plants are being connected to local electric-supply systems every day.

However, according to [1], this causes instability in the grid, which is one of the greatest challenges to the energy industry. Electrical operators need to know how much energy will be added to the system in order to balance it with consumption and ensure that the system is capable of meeting consumer demand. For [2], the ability to predict photovoltaic solar power output is very important for secure grid operation, scheduling, and the effectiveness of power-grid management.

In this context, statistical models emerge as important tools for modeling and predicting photovoltaic power generation. Some statistical approaches used for modeling the solar photovoltaic power generation include linear regression models [3,4,5,6], autoregressive models [7,8,9], and artificial-neural-network models [10,11,12], that is, parametric models are still commonly employed due to their ease of use.

However, parametric models have at least three limitations: (i) the analysis is limited to the function (or functions) previously chosen by the analyst; (ii) the complexity and/or flexibility of the functions considered is limited by the number of parameters in the functions; and (iii) there may exist several functions that can fit the recorded values equally well. A common solution adopted in much statistical analysis is to fit a set of candidate models and then choose the best model using some model-selection criterion, such as AIC [13] or BIC [14]. Even in those cases, issues (ii) and (iii) still remain.

In this paper, in order to give models more flexibility instead of restricting them to a function f(·) chosen previously, we adopt a semi-parametric Bayesian approach, in which the curve of the function f(·) is estimated from the observed data. For this, we assume that cumulative solar photovoltaic power generation, measured at *k* time instants per day, is modeled by an additive model composed of a nonlinear growth function f(·) evaluated at *k* time instants plus a random error ε. However, instead of setting up f(·) as a known mathematical function, we assume that f(·) is an unknown function whose vector of values $f\left(t\right)=\left(f(1),\ldots ,f(k)\right)$ is treated as a set of parameters that must be estimated based on recorded data for t=(1,…,k). To jointly estimate f(t) and the other parameters in the proposed model, we adopt a Bayesian approach with a Gaussian-process prior over f(t). An advantage of this approach is that we may estimate the curve of function f(·) using “smooth functions” that are obtained by linking points generated from a *k*-variate normal distribution with an appropriate mean vector and covariance matrix. Additionally, we present a forecasting procedure for the value of the curve on the (n+1)th day, conditioned on the values recorded over the first *n* days.

Since the joint posterior distribution for the parameters and the predictive distribution of the proposed model do not have known mathematical forms, we describe how to implement a Gibbs sampling algorithm [15,16,17] to generate random values from these two distributions. This algorithm generates values from the distributions of interest in an indirect way, using the conditional posterior distributions, as long as those are known, which is the case for the model proposed here.

To illustrate the performance of the proposed model, we include two simulation studies. In the first one, we examine the performance of the proposed model in the estimation of the curve of f(·). As performance measures, we calculate the absolute percentage error (APE) and the mean absolute percentage error (MAPE). In all simulated cases, the proposed approach presents MAPE values near 0, indicating that the estimated values are close to the real values. In the second simulation study, we evaluate the performance of predictions made with the proposed approach. Analogously to the first simulation study, the proposed approach presents satisfactory performance, as indicated by MAPE values near zero. In addition to APE and MAPE values, we also evaluate the performance of predictions in terms of the root-mean-square error (RMSE). The RMSE values were all near zero, indicating a very good performance of the proposed approach. We also apply the proposed approach to a real dataset. Like in the simulation studies, the results obtained in this application were very accurate, with MAPE and RMSE values near zero.

The main novelty that we bring in this paper is in the way that we model the generation of solar photovoltaic power over time. First, we consider the photovoltaic power generated on each day as having its own behavior, and the behavior is taken to be proportional to the average behavior of the measurements over the last *n* days. Second, the function f(·) that models the generation of solar power as a function of time is considered unknown, but with its curve estimated from the observed data. We highlight the following four advantages of this approach: (i) the proposed hierarchical Bayesian model is very flexible and adapts to the number of values recorded; (ii) the inference procedure is based on a Gibbs sampling algorithm, which can be easily implemented in statistical software such as R; (iii) the predicted growth curve for day (n+1) is obtained directly using only the history of the first *n* measurements; and (iv) there is no need to fit a set of models and afterwards compare them using some model-selection criterion.

The remainder of the paper is organized as follows. In Section 2, we present the dataset that has motivated us to develop the proposed modeling, the hierarchical Bayesian model, and the estimation procedure for the parameters of interest. In Section 2, we also present the results of the first simulation study. In Section 3, we present the prediction procedure and the second simulation study. In Section 4, we apply the proposed approach to a real dataset. Finally, in Section 5, we conclude with some final remarks.

## 2. Dataset and Statistical Modeling

A critical component of any statistical analysis is the dataset used to make inferences on the parameters of interest. The dataset used in this paper was obtained from a photovoltaic plant installed on the campus of the Brazilian Federal University of Mato Grosso du Sul. This dataset is freely available on the website https://github.com/lscad-facom-ufms/Solar2 (accessed on 3 June 2024), and more details on the experiment can be found in [18]. The dataset used to make inferences on the parameters of the proposed model contains measurements of solar photovoltaic power generation taken at k=74 time instants each day over a period of N=19 days. In other words, the dataset is a spreadsheet composed of 3 columns and k×n=74×19=1406 lines. Figure 1 shows a clipping from the data spreadsheet, showing that the first column contains the day (1–19), the second column the time instant (TI, 1-74), and the third column the observed values for photovoltaic solar power (PSP) generated.

Let $W_{it}$ be the solar power recorded at the *i*th time instant of the *i*th day and $W_{i}=\left(W_{i1},\ldots ,W_{ik}\right)$ the vector of values recorded on the *i*th day, for i=1,…,N and t=1,…,k. Figure 2 shows the values recorded over the first two days of the experiment. As one can note, the recorded data on these two days present great variability, which makes the modeling process difficult. For the other days, the recorded values present similar behavior. Due to this, we opt to model the accumulated values.

Consider $W_{it}^{\mathrm{ac}}=\sum_{t^{\prime }=1}^{t}W_{it^{\prime }}$ to be the accumulated values of the photovoltaic power generated through the *t*th time instant of the *i*th day and $W_{i}^{ac}=\left(W_{it}^{ac},\ldots ,W_{ik}^{ac}\right)$ to be the vector of accumulated values, for i=1,…,n and t=1,…,k. Figure 3a shows the accumulated values recorded over the first four days of the experiment. As one can note, the accumulated values present more stable and predictable behavior. However, many values in the vectors $W_{i}^{ac}$ are on the scale of 100,000, which could cause computational problems in the inference process. To avoid this problem, we opt to model the *logarithm* of the accumulated values, denoted by $Y_{it}=\log\left(W_{it}^{ac}\right)$, for i=1,…,N and t=1,…,k. Additionally, let $Y_{i}=\left(Y_{i1},\ldots ,Y_{ik}\right)$ be the vector of values recorded on the *i*th day, for i=1,…,N.

Figure 3b, shows the graph of Y values recorded over the first four days of the experiment. The symbols • in black connected by black lines represent the average values $\bar{y}=\left({\bar{y}}_{1},\ldots ,{\bar{y}}_{k}\right)$. For the other days, the recorded Y values present similar behavior. From this point forward, and without loss of generality, consider the modeling of $y=\left(y_{1},\ldots ,y_{n}\right)$, that is, the data recorded over the first *n* days of the experiment, for n<N, where the primary interest is in in the prediction of the values that will be generated on day (n+1). That is, y is an n×k matrix in which row *i* contains the *y* values generated on day i, for i=1,…,n.

### 2.1. Hierarchical Bayesian Model

Based on Figure 3b, consider the average curve (black line) to be a nonlinear growth function f(t), where $f\left(t\right)=\left(f(1),\ldots ,f(k)\right)$ is a vector composed of the values of f(·) at *k* instants of time, t=(1,…,k). Assume that the growth function for the *i*th day is proportional to f(·), i.e., $f_{i}\left(\cdot \right)=C_{i}f\left(\cdot \right)$, for $C_{i}>0$ and i=1,…,n with n≤N. In other words, we are assuming that there is a growth function f(·) whose curve represents the average curve from *n* curves, with the curve on the *i*th day proportional to the average curve.

Consider recorded values on the *i*th day to be generated according to the following additive model:(1)$$Y_{i}=\left(Y_{i1},\ldots ,Y_{ik}\right)∼C_{i}\cdot f\left(t\right)+\varepsilon_{i},$$
with $C_{i}>0$, where $\varepsilon_{i}=\left(\varepsilon_{i1},\ldots ,\varepsilon_{ik}\right)$ is a vector of random errors, for i=1,…,n. Assume that $\varepsilon_{i}$ is generated according to a *k*-dimensional multivariate normal distribution with mean vector 0=(0,…,0) and covariance matrix Σ (dimension k×k) composed of the elements $\sigma_{\varepsilon }\left(t,t^{\prime }\right)=Cov\left(\varepsilon_{it},\varepsilon_{it^{\prime }}\right)$, for $t,t^{\prime }=1,\ldots ,k$ and i=1,…,n.

To complete Model (Equation 1), we could fix f(·) as a known mathematical function, such as the logistic or Gompertz growth functions, among others. However, three problems with this parametric approach are (i) the analysis is limited to the function (or functions) previously chosen by the analyst; (ii) the complexity and/or flexibility of the considered functions is limited by the number of parameters in the functions; and (iii) there may exist several functions that can fit the recorded values equally well. A common solution to problem (i) is to fit a set of candidate models and then choose the best model using some model selection criterion, such as AIC [13] or BIC [14]. However, issues (ii) and (iii) still remain.

In order to avoid restricting our model to a specific parametric function, from this point onward, we assume that f(·) is an unknown function and that the values $f\left(t\right)=\left(f(1),\ldots ,f(k)\right)$ are model parameters that need to be estimated from observed data. Under this scenario and with the model given by (Equation 1), the parameters of interest are $\theta =\left(f(t),\Sigma ,C\right)$, where $f\left(t\right)=\left(f(1),\ldots ,f(k)\right)$, Σ is the covariance matrix of the vector of random errors, and $C=(C_{1},\ldots ,C_{n})$.

To estimate θ, we take a hierarchical Bayesian approach with a Gaussian-process prior on f(t), denoted by $f\left(t\right)|m,\Sigma_{f}∼\mathrm{GP}\left(m,\Sigma_{f}\right)$. This means that we are considering f(·) as an unknown function, but with the vector of values $f\left(t\right)=\left(f(1),\ldots ,f(k)\right)$ generated by a *k*-variate normal distribution with mean vector m and covariance matrix $\Sigma_{f}$ composed of the elements $\sigma_{f}\left(t,t^{\prime }\right)=C\mathrm{ov}\left(f\left(t\right),f\left(t^{\prime }\right)\right)$, for $t,t^{\prime }=1,\ldots ,k$. For Σ, we assume a conjugated inverse-Wishart prior distribution with parameter $\left(\delta ,V\right)$, and, for $C_{i}$, we assume a prior distribution given by a truncated normal distribution (with the left-of-zero part removed) with parameters $\mu_{c}$ and $\sigma_{c}^{2}$, for i=1,…,n. The proposed model is then represented hierarchically:(2)$$\begin{array}{ccc} Y_{i}=\left(Y_{i1},\ldots ,Y_{ik}\right)|f\left(t\right),\Sigma ,C & ∼ & N_{k}\left(C_{i}f\left(t\right),\Sigma \right)\\ f\left(t\right)|m,\Sigma_{f} & ∼ & \mathrm{GP}\left(m,\Sigma_{f}\right)\\ \Sigma |\delta ,V & ∼ & \mathrm{IW}\left(\delta ,V\right)\\ C_{i}|\mu_{c},\sigma_{c}^{2} & ∼ & N_{\mathrm{trunc}}\left(0;\mu_{c},\sigma_{c}^{2}\right), \end{array}$$
where $N_{k}\left(\cdot \right)$, GP(·), IW(·), and $N_{\mathrm{trunc}}\left(0;\cdot \right)$ represent, respectively, a *k*-variate Gaussian distribution, the Gaussian process, the inverse-Wishart distribution, and the truncated normal distribution with values only on the right half-line; additionally, m, $\Sigma_{f}$, δ, V, $\mu_{c}$, and $\sigma_{c}^{2}$ are known hyperparameters, for i=1,…,n.

We complete the modeling by setting the following:(i)m=0 in order to represent our lack of informative prior knowledge about the expected value of f(t);(ii)$\Sigma_{f}=\lambda W$, with λ>0 and W a matrix of dimension k×k composed of elements $\kappa (t,t^{\prime })$, calculated according to the squared exponential kernel, i.e.,
(3)$$\kappa \left(t,t^{\prime }\right)=\eta^{2}\exp\left\{-\frac{{\left(t-t^{\prime }\right)}^{2}}{2\nu^{2}}\right\},$$
with η,ν>0. The parameter η controls how far the generated values are from the average. Small values for η characterize functions that are close to their average value, whereas larger values allow greater variation. The parameter ν controls the smoothness of the function obtained by connecting the points. Small values of ν mean that function values may change quickly, and large values characterize functions that change more slowly (are smoother). We set λ=100, η=1, and ν=1 in order to obtain a weakly informative prior distribution;(iii)We finalize the model by setting δ=k, $V=0.01\cdot I_{k}$, where I is the identity matrix of dimension k×k, and $\mu_{c}=\sigma_{c}^{2}=1$.

Applying Bayes’s theorem, the joint posterior distribution for $\theta =\left(f(t),\Sigma ,C\right)$ is given by
(4)$$\pi \left(\theta |y,t\right)\propto L\left(\theta |y,t\right)\pi \left(f\left(t\right)|m,\Sigma_{f}\right)\pi \left(\Sigma |\delta ,V\right)\pi \left(C|\mu_{c},\sigma_{c}^{2}\right),$$
where L(θ|y,t) is the likelihood function of a *k*-variate normal distribution with parameters f(t) and Σ, and π(·) represents the probability density functions of the prior distributions, for $y=(y_{1},\ldots ,y_{n})$ and t=(1,…,k).

However, the joint posterior distribution does not have a known mathematical form that allows us to generate random values from this distribution directly. Due to this, we need to use an algorithm that generates the random numbers of this joint distribution in an indirect way. In this paper, we opt to generate random values from π(θ|y,t) using the Gibbs sampling algorithm [15,17] due to its simplicity of implementation and efficiency. This algorithm generates values from the joint posterior distribution indirectly, using the conditional posterior distributions, as long as they are known.

For the proposed hierarchical Bayesian model, the conditional posterior distributions are known and given by
(5)$$\begin{array}{ccc} f(t)|y,t,• & ∼ & \mathrm{GP}\left(\Sigma^{-1}{\left(\sum_{i=1}^{n}C_{i}\Sigma^{-1}+\Sigma_{f}^{-1}\right)}^{-1}\sum_{i=1}^{n}C_{i}y_{i},{\left(\sum_{i=1}^{n}C_{i}\Sigma^{-1}+\Sigma_{f}^{-1}\right)}^{-1}\right) \end{array}$$
(6)$$\begin{array}{ccc} \Sigma |y,t,• & ∼ & \mathrm{IW}\left(\delta +k+n,V+\sum_{i=1}^{n}{\left(y_{i}-C_{i}f\left(t\right)\right)}^{⊤}\left(y_{i}-C_{i}f\left(t\right)\right)\right) \end{array}$$
(7)$$\begin{array}{ccc} C_{i}|y,t,• & ∼ & N_{\mathrm{trunc}}\left(0,\frac{f{\left(t\right)}^{⊤}\Sigma^{-1}y_{i}+1)}{f{\left(t\right)}^{⊤}\Sigma^{-1}f\left(t\right)+1},\frac{\sigma_{c}^{2}}{f{\left(t\right)}^{⊤}\Sigma^{-1}f\left(t\right)+\mu }\right), \end{array}$$
where the symbol • represents all other parameters.

Using the conditional posterior distributions, we implement a Gibbs sampling algorithm according to the steps described in Algorithm 1.
**Algorithm 1** Gibbs sampling algorithm.1:Let the state of the Markov chain consist of $\theta =\left(f(t),\Sigma ,C\right)$.2:Initialize the algorithm with a configuration $\theta^{\left(0\right)}=\left(f{\left(t\right)}^{\left(0\right)},\Sigma^{\left(0\right)},C^{\left(0\right)}\right)$.3:**procedure** For the *l*th iteration of the algorithm, l=1,…,L:  4:    generate $f{\left(t\right)}^{\left(l\right)}$ from conditional distribution (Equation 5), given $\Sigma^{\left(l-1\right)}$ and $C^{\left(l-1\right)}$;   5:    generate $\Sigma^{\left(l\right)}$ from conditional distribution (6), given $f{\left(t\right)}^{\left(l\right)}$ and $C^{\left(l-1\right)}$;   6:    generate $C_{i}^{\left(l\right)}$ from conditional distribution (7), given $f{\left(t\right)}^{\left(l\right)}$ and $\Sigma^{\left(l\right)}$, for i=1,…,n.

After running *L* iterations of the Gibbs sampling algorithm, we discard the first *B* iterations as a burn-in. We also consider jumps of size *J*, i.e., only 1 drawn from every *J* was extracted from the original sequence in order to obtain a sub-sequence of size S=[(L−B)/J] to make inferences. The estimates for the parameters of interest are given by the average of the generated values, i.e.,
$$\hat{f}\left(t\right)=\frac{1}{S}\sum_{l=1}^{S}f{\left(t\right)}^{\left(M(l)\right)},\;\;\hat{\Sigma }=\frac{1}{S}\sum_{l=1}^{S}\Sigma^{\left(M(l)\right)}\;\;\mathrm{and}\;\;{\hat{C}}_{i}=\frac{1}{S}\sum_{l=1}^{S}C_{i}^{\left(M(l)\right)}$$
where $f{\left(t\right)}^{\left(M(l)\right)}$, $\Sigma^{\left(M(l)\right)}$, and $C^{\left(M(l)\right)}$ are the generated values for parameters f(t), Σ, and C, respectively, in the M(l)=(B+1+(l−1)·J)th iteration of the algorithm, for l=1,…,S. The 95% credibility interval for each of the parameters is given by the 2.5% and 97.5% percentiles of the sampled values. The estimated curve of f(·) is obtained by plotting the points $\left(t,\hat{f}\left(t\right)\right)$ connected by lines, for t=1,…,k.

### 2.2. First Simulation Study

To illustrate the performance of the proposed approach, we develop a simulation study. To generate the dataset, we consider f(·) as the log-Gompertz function of parameters $\alpha_{1}$, $\alpha_{2}$, and $\alpha_{3}$ with the parametrization $f\left(t\right)=\log\left(\alpha_{1}\right)-\exp\left\{\alpha_{2}-\alpha_{3}t\right\}$, for t>0. We set $\alpha_{1}=12$, $\alpha_{2}=2$, and $\alpha_{3}=0.1$. For the covariance matrix Σ, we calculate each term according to the squared-exponential kernel given in Equation (Equation 3) with $\eta^{2}=0.01$ and $\nu^{2}=10$.

The procedure to generate the artificial dataset is given by the following four steps:(i)Fix the number of days *n* and the number of time instants per day *k*;(ii)With t=(1,…,k), calculate $f\left(t\right)=\left(f(1),\ldots ,f(k)\right)$, where f(t) is given by the log-Gompertz function described above;(iii)Fix the values $C=(C_{1},\ldots ,C_{n})$, with $C_{i}>0$ and i=1,…,n;(iv)Generate $Y_{i}=\left(Y_{i1},\ldots ,Y_{ik}\right)∼N_{k}\left(f{\left(t\right)}^{⊤},\Sigma \right)$, for i=1,…,n

To simplify the visualization of the results, our first simulation study considers n=4 and k=50. We set $C=\left(C_{1},C_{2},C_{3},C_{4}\right)=\left(0.8,0.9,1.1,1.2\right)$. Figure 4a shows the curve of f(t), which we call the “average curve”, and the curve for the *i*th day is given by $C_{i}f\left(t\right)$, for i=1,2,3,4. Figure 4b shows the curves and actual y values generated for each day (coloured • symbols), in which black • symbols are the average values $\bar{y}$.

With the dataset generated, we apply the proposed Gibbs sampling algorithm with *L* = 55,000 iterations, B=5000, and J=10. In this way, we obtain a posterior sample of size S=5000 to make inferences. To verify how far the estimates $\hat{f}\left(t\right)$ are from the real values f(t), we calculate the absolute percentage error:$$\mathrm{APE}\left(d_{t}\right)=\frac{|f\left(t\right)-\hat{f}\left(t\right)|}{f(t)}\cdot 100,$$
for t=1,…,k.

Figure 5a shows f(t) (black line) and the estimated curve by the proposed method (red line) with a credibility band of 95% (red region). Figure 5b shows $\mathrm{APE}\left(d\right)=(\mathrm{APE}\left(d_{1}\right),\ldots ,$
$\mathrm{APE}(d_{k}))$. $\mathrm{APE}\left(d\right)$ values ranged from a minimum of 0.0145 to a maximum of 1.4481, with a mean absolute percentage error (MAPE) of 0.4568. The small values of APE indicate that the estimated values $\hat{f}\left(t\right)$ are very close to the real values of f(t), for t=1,…,k.

Similarly, we obtained the estimated curve for the *i*th day by plotting the pairs $\left(t,{\hat{y}}_{it}\right)$ connected by lines, for ${\hat{y}}_{it}=\hat{C_{i}}\hat{f}\left(t\right)$, i=1,…,n, and t=1,…,k. For this case, we calculate two different kinds of error: the APE for comparison between the real value $C_{i}f\left(t\right)$ and the estimated value ${\hat{y}}_{it}$; and the APE for comparison between the generated values $y_{it}$ and the estimated ${\hat{y}}_{it}$ values. They are calculated as follows:$$\mathrm{APE}\left(d_{it}\right)=\frac{|C_{i}f\left(t\right)-{\hat{y}}_{it}|}{C_{i}f\left(t\right)}\cdot 100\;\;\;\;\;\;\mathrm{and}\;\;\;\;\;\;\mathrm{APE}\left(e_{it}\right)=\frac{|y_{it}-{\hat{y}}_{it}|}{y_{it}}\cdot 100$$
for i=1,…,n and t=1,…,k.

Table 1 shows the estimates and 95% credibility intervals for parameters $C_{i}$, i=1,2,3,4. As one can note, the estimated values are very close to the real values, and the real values are inside the credibility intervals.

Figure 6a shows $C_{i}f\left(t\right)$ (black line) and the estimated curves by the proposed method (red lines), where the symbols • are the generated values for each day. Figure 6b shows the values of $\mathrm{APE}\left(d_{i}\right)$, and Table 2 shows the summary measures of $\mathrm{APE}(d_{i})$ values, for i=1,…,n. As one can note, the estimated curves for each one of the four days is satisfactorily close to the real curves, as indicated by $\mathrm{APE}(d_{i})$ values near zero, i=1,…,n.

Figure 7 shows the graphic of the values of $\mathrm{APE}\left(e_{i}\right)=\left(\mathrm{APE}\left(e_{1}\right),\ldots ,\mathrm{APE}\left(e_{k}\right)\right)$, and Table 3 shows the summary measures of $\mathrm{APE}(e_{i})$ values, for i=1,…,n. $\mathrm{APE}(e_{i})$ values are near zero, indicating that the estimated values ${\hat{y}}_{it}$ are satisfactorily close to the generated values $y_{it}$, for i=1,…,n and t=1,…,k.

Since the inferences were made from a posterior sample obtained from an MCMC algorithm, it is important to check the convergence of the sampled values. As is usual, we verify the convergence of the sampled values empirically, using the ergodic mean (ErM) of the sampled values. Figure 8 shows the graphic of the ErM for the sampled values for f(1) and f(30). As one can note, there is no reason to doubt the convergence of the sampled values since the ErM values present satisfactory stabilization. The graphs of ErM for the sampled values for other parameters are similar.

## 3. Predictions

In addition to obtaining the estimates $\hat{\theta }=\left(\hat{f}\left(t\right),\hat{\Sigma },\hat{C}\right)$ for the parameters $\theta =\left(f(t),\Sigma ,C\right)$, the modeling used in the previous section may also be used to predict the values that will be recorded on the (n+1)th day, i.e., $Y_{n+1}=\left(Y_{1(n+1}),\ldots ,Y_{k(n+1)}\right)$. This can be performed by using the predictive distribution, given by
(8)$$\begin{array}{c} \pi \left(Y_{n+1}|y,t\right)=\int \pi \left(Y_{n+1}|y,t,\theta \right)\pi \left(\theta |y,t\right)d\theta , \end{array}$$
where $\pi \left(\theta |y,t\right)$ is the joint posterior distribution for θ, given in Equation (Equation 4). However, this integral does not have a known analytic solution. Due to this, we present in the following an MCMC algorithm for obtaining an approximation for this integral.

From Model (Equation 1), the marginal distribution for $Y_{i}$ is given by a *k*-variate normal distribution with mean vector $0={\left(0,\ldots ,0\right)}^{⊤}$ and covariance matrix $C_{i}^{2}\Sigma_{f}+\Sigma$, for i=1,…,n. Thus,
$$Y=\left(Y_{1},\ldots ,Y_{n}\right)|\Sigma_{y}∼N_{nk}\left(0,\Sigma_{y}\right),$$
where $N_{nk}\left(\cdot \right)$ represents an nk-variate normal distribution with mean vector 0 and a covariance matrix of dimension nk×nk, given by
$$\Sigma_{y}=\left[\begin{array}{ccccc} C_{1}^{2}\Sigma_{f}+\Sigma & \Sigma_{12} & \Sigma_{13} & \ldots & \Sigma_{1n}\\ \Sigma_{21} & C_{2}^{2}\Sigma_{f}+\Sigma & \Sigma_{23} & \ldots & \Sigma_{2n}\\ \vdots & \vdots & \vdots & \vdots \;\vdots \;\vdots & \vdots \\ \Sigma_{n1} & \Sigma_{n2} & \Sigma_{n3} & \ldots & C_{n}^{2}\Sigma_{f}+\Sigma \end{array}\right],$$
where $\Sigma_{ii^{\prime }}=C_{i}C_{i^{\prime }}\Sigma_{f}$ are the covariance matrices (of dimension k×k) among the measurements $Y_{i}$ and $Y_{i^{\prime }}$, for $i,i^{\prime }=1,\ldots ,n$ and $i\neq i^{\prime }$. Similarly, $Y_{n+1}∼N_{k}\left(0,C_{n+1}^{2}\Sigma_{f}+\Sigma \right)$. Therefore, the joint distribution for $\left(Y,Y_{n+1}\right)$ is
$$\left[\begin{array}{c} Y\\ Y_{n+1} \end{array}\right]|\theta ,y,t∼N_{(n+1)k}\left(\left[\begin{array}{c} 0\\ 0 \end{array}\right],\left[\begin{array}{cc} \Sigma_{y} & B^{T}\\ B & C_{n+1}^{2}\Sigma_{f}+\Sigma \end{array}\right]\right),$$
where $B=\left[\Sigma_{(n+1)1}\;\Sigma_{(n+1)2}\;\ldots \;\Sigma_{(n+1)n}\right]$ is a block of matrices, in which $\Sigma_{(n+1)i}$ are the covariance matrices among the values of $Y_{n+1}$ and $Y_{i}$ given by $\Sigma_{(n+1)i}=C_{n+1}C_{i}\Sigma_{f}$, for i=1,…,n.

Using the properties of the multivariate normal distribution, the conditional posterior distribution for $Y_{n+1}$ is given by
(9)$$Y_{n+1}|y,t,\theta ∼N_{k}\left(B\Sigma_{y}^{-1}y,\left(C_{n+1}^{2}\Sigma_{f}+\Sigma \right)-B^{⊤}{\left(C_{n+1}^{2}\Sigma_{f}+\Sigma \right)}^{-1}B\right);$$
with $C_{n+1}$ generated from
(10)$$C_{n+1}|C∼Ntrunc\left(0,\bar{C},S_{c}^{2}\right),$$
where $\bar{C}$ and $S_{c}^{2}$ are, respectively, the average and the variance of $C=(C_{1},\ldots ,C_{n})$.

Thus, a sample from the conditional posterior distribution of $\left(Y_{n+1},\theta \right)$ can be generated according to the steps in Algorithm 2.
**Algorithm 2** Prediction.1:Let the state of the Markov chain consist of $\theta =\left(f(t),\Sigma ,C\right)$ and $Y_{n+1}$.2:Initialize the algorithm with a configuration $\theta^{\left(0\right)}=\left(f{\left(t\right)}^{\left(0\right)},\Sigma^{\left(0\right)},C^{\left(0\right)}\right)$ and $C_{n+1}^{\left(0\right)}$.3:**procedure** For the *l*th iteration of the algorithm, l=1,…,L:  4:    Update θ according to Algorithm 1;   5:    Generate $Y_{n+1}^{\left(l\right)}$ from conditional posterior distribution in (Equation 9) given $C_{n+1}^{\left(l-1\right)}$;   6:    Generate $C_{n+1}$ from probability distribution in (Equation 10).

After running the algorithm for the same *L* iterations, burn-in *B*, and jump *J* as we used for algorithm (1), an approximation for the integral in (Equation 8) is given by
$$\tilde{\pi }\left(Y_{n+1}|y\right)=\frac{1}{S}\sum_{l=1}^{L}Y_{n+1}^{\left(M(l)\right)},$$
where M(l) is the (B+1+l·J)th iteration of the algorithm, for l=1,…,S.

### Second Simulation Study

To illustrate the performance of the prediction procedure, we present a second simulation study. Like in the first simulation study, we fix n=4, k=50, and f(t) as the log-Gompertz function of parameters $\alpha_{1}=12$, $\alpha_{2}=2$, and $\alpha_{3}=0.1$. Here, the main objective is to predict the curve for the (n+1) = 5th day.

To obtain the curves of the first four days with the curve of f(t) being the average curve, we adopt the following procedure:(i)Let $C_{i}<n$ and define $p=\left(p_{1},p_{2},p_{3},p_{4}\right)$ with $p_{i}=\frac{C_{i}}{n}$, for i=1,2,3,4;(ii)Generate p∼Dirichlet(α), where Dirichlet(α) is the Dirichlet distribution with parameter α. We set up α=(50,50,50,50);(iii)Obtain $C_{i}=n\cdot p_{i}$ and generate $Y_{i}=\left(Y_{i1},\ldots ,Y_{ik}\right)∼N_{k}\left(C_{i}f\left(t\right),\Sigma \right)$, for i=1,…,n, where Σ is obtained as described in simulation study 1.

The generated values for $C=\left(C_{1},C_{2},C_{3},C_{4}\right)$ were (0.9187,0.8767,1.0126,1.1919), respectively. Figure 9a shows the real curves for days 1 to 4, denoted by $C_{i}f\left(t\right)$ for i=1,2,3,4, and Figure 9b shows the same graphs as Figure 9a with the generated values for each day as correspondingly coloured • symbols, the black • symbols being the average values.

With $\left(Y_{1},\ldots ,Y_{n}\right)$ values generated, we generate the data for the (n+1)th day as follows:(i)Generate $C_{n+1}∼N_{\mathrm{trunc}}\left(0,\bar{C},S_{c}^{2}\right)$, where $\bar{C}=\frac{1}{n}\sum_{i=1}^{n}C_{i}$ and $S_{c}^{2}=\frac{1}{n-1}\sum_{i=1}^{n}{\left(C_{i}-\bar{C}\right)}^{2}$. From the generated C=(0.9187,0.8767,1.0126,1.1919) values, the generated value for $C_{n+1}$ was 0.9823;(ii)Generate $Y_{n+1}=\left(Y_{(n+1)1},\ldots ,Y_{(n+1)k}\right)∼N_{k}\left(C_{n+1}f\left(t\right),\Sigma \right)$.

We then use the generated values for $\left(Y_{1},\ldots ,Y_{n}\right)$ in the the prediction procedure (Algorithm 2) and obtain the estimates for ${\hat{Y}}_{n+1}$. For this, we apply the prediction procedure for the same *L* = 55,000 iterations with a burn-in B=5000 and jump of size J=10 as we used in Algorithm 1. The estimated curve for the (n+1)th day is obtained by plotting the pairs $\left(t,{\hat{y}}_{(n+1)t}\right)$ connected by lines, where, ${\hat{y}}_{(n+1)t}$ is the predicted value for $Y_{(n+1)t}$, for t=1,…,k.

Figure 10a shows f(t) and the estimated curve by the proposed method, and Figure 10b shows the graph of APE(d) values. As in the first simulation study, the results show a very satisfactory performance of the proposed method, with the estimated curve very close to the real curve of f(t), as indicated by APE(d) values all being less than 1.

Figure 11a shows $C_{i}f\left(t\right)$ (black line) and the estimated curves by the proposed method (red lines), with the symbols • representing the generated values for each day. Figure 11b shows $\mathrm{APE}\left(d_{i}\right)$, for i=1,…,n. As one can note, the estimated curves for each one of the four days are satisfactorily near the real curves, as indicated by $\mathrm{APE}(d_{i})$ values near zero, for i=1,2,3,4. We also verify the convergence of the sampled values. Analogously to results presented in the first simulation study, there is no reason to doubt the convergence of the sampled values since the ErM values present satisfactory stabilization.

Figure 12a shows the curve for the (n+1)th day (green line), with the green • symbols representing the generated data for the (n+1)th day, the predicted curve (red line), and a posterior prediction 95% credibility band (red region). The estimate for $C_{n+1}$ is ${\hat{C}}_{n+1}$ = 0.9936 with a 95% credibility interval given by (0.7207,1.2716), that is, the real value $C_{i}=0.9823$ is inside the credibility interval. Additionally, the real curve is completely inside the 95% posterior prediction band. Figure 12b shows the graph of APE(d) and APE(e) in relation to predicted values. All APE values are smaller than 3, indicating that predicted values are close to real values $C_{n+1}f\left(t\right)$ and to the generated values for $Y_{n+1}$.

In addition to APE values, we also calculate the root-mean-square error (RMSE) in order to have one more performance measure of the predictions, given by
$$RMSE=\sqrt{\frac{1}{k}\sum_{t=1}^{k}{\left(y_{(n+1)t}-{\hat{y}}_{(n+1)t}\right)}^{2}},$$
where $y_{(n+1)t}$ is the value generated for the *t*-th time instant of the (n+1)-th day, and ${\hat{y}}_{(n+1)t}$ is the respective predicted value, for t=1,…,k. The RMSE value obtained was 0.2021, that is, similar to the APE values, the RMSE value also indicates that the predicted values are satisfactorily close to the generated values.

In order to avoid restricting the model to the results of just one artificial dataset, we repeat the second simulation study M=100 times and calculate the percentage of times that the real curve for the (n+1)th day is completely inside the prediction band of 95% and the average of the APE and RMSE values. Overall, in 96% of simulated cases, the real curve of $C_{n+1}f\left(t\right)$ is completely inside the posterior prediction band, the average of the MAPE values is 0.9550, and the average of the RMSE values is 0.1534. Figure 13 shows the APE and RMSE values for the M=100 simulations. Note that both results show a very satisfactory performance of the proposed method. As an illustration of the predictions results, Figure 14 shows the predicted curve with a 95% posterior prediction band (red region) and the real curve for the 18th and 27th simulations. For these two simulations, the MAPE values were 0.6115 and 4.0017, respectively; and the RMSE values were 0.0622 and 0.4996, respectively.

## 4. Application

We now apply the proposed approach to the real dataset described in Section 2. For this application, we use the same hyperparameter values and the same *L*, *B*, and *J* values used in the two simulation studies. Additionally, we use n=4, i.e., we apply the proposed approach for estimating the curve of f(t) using the dataset of four days, and then we predict the curve for the (n+1)th day. This procedure was applied for the data recorded over the first 19 days of the experiment, always using a window of 4 days, to obtain the estimated curve of f(t) and to predict the curve for the (n+1)th day. Thus, overall, 15 analyses were carried out with predictions for days 5 to 19.

Our first application considers the recorded data on the first four days of the experiment to estimate the curve of f(t), and then we predict the curve for the fifth day (n+1=5). Figure 15a shows the average values recorded in the first four days (symbols •), the estimated curve of f(t) (red line) and a 95% credibility band (red region). Figure 15b shows APE values for comparison between $\bar{y}$ and the estimated values $\hat{y}=\left({\hat{y}}_{1},\ldots ,{\hat{y}}_{n}\right)$. The MAPE value is 0.2590. These results shows that the estimated values are very close to the recorded average values. In other words, the proposed approach presented a very satisfactory performance in estimating the average values recorded over the first four days on which the experiment was carried out.

Table 4 shows MAPE and RMSE values for the 15 analyses. MAPE values range from a minimum of 0.0095 for day 13 to a maximum of 0.5360 for day 11, with an average value of 0.2349. RMSE values range from 0.0011 for day 13 to a maximum of 0.0612 for day 11, with an average value of 0.0265. As one can see, all MAPE and RMSE values are near zero, indicating that the estimated values $\hat{y}$ are close to the recorded $\bar{y}$ for the 15 analyses.

Table 5 shows MAPE and RMSE values for the 15 predictions. As one can note, MAPE values vary from a minimum of 0.3344 for day 19 to a maximum of 7.1648 for day 13. The average value of MAPE over the 15 days was 2.5719. Similarly, RMSE values range from 0.0382 for day 19 to a maximum of 0.7410 for day 13, with an average value of 0.2895. Overall, these results show that the predicted values ${\hat{y}}_{i}^{pred}$ are close to the recorded values $y_{i}$, for i=5,…,19.

As an illustration of the good performance of the proposed approach in predictions, Figure 16 shows the recorded values (symbols •), the predicted curve (red line), and a 95% prediction credibility band (red region) for the recorded values on days 9 and 19, which are the two days with the smallest MAPE and RMSE values. Figure 17 shows the prediction results for days 13 and 14, which are the two days with the greatest MAPE and RMSE values. Although predictions for days 13 and 14 present the two highest MAPE values, most of the recorded values are inside the 95% prediction credibility band. Overall, the proposed approach presented very satisfactory performance, as indicated by the small MAPE and RMSE values.

## 5. Final Remarks

In this paper, we propose a Bayesian approach for modeling and forecasting photovoltaic solar power generation. For this, we assume that the growth curve of the generated power over the time of day is proportional to an average curve whose associated function is denoted by f(t). However, instead of taking a parametric approach by setting up f(t) as a known mathematical function, we assume that f(t) is an unknown function, but with the vector of values $f\left(t\right)=\left(f(1),\ldots ,f(k)\right)$ generated a priori from a Gaussian process. To perform inference for the parameters of interest θ, we use a Gibbs sampling algorithm.

The good performance of the proposed approach and its four advantages as described in Section 1 were illustrated by means of two simulation studies and an application to a real dataset. The results obtained show that the proposed approach is an efficient alternative for modeling the solar power generated on the days considered in the study and also for forecasting next-day solar power generation.

From a practical point of view, the results show that the proposed modeling and the estimation procedure were very accurate in predicting energy generation for the next day. Although the proposed approach has been described as forecasting the growth curve for the next day, it can also be used to forecast power generation for short intervals, such as hourly power generation. An extension of the approach presented here is the inclusion of explanatory variables in the modeling since power generation is influenced by environmental variables such as temperature and irradiance. All computational implementations were carried out using the R software [19], and the code can be obtained by e-mailing the authors.

## Figures and Tables

**Figure 1 entropy-26-00824-f001:**
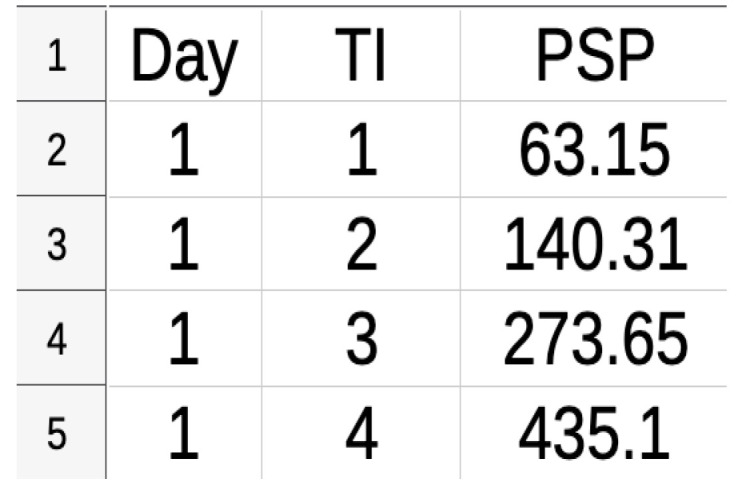
Clipping of the data spreadsheet.

**Figure 2 entropy-26-00824-f002:**
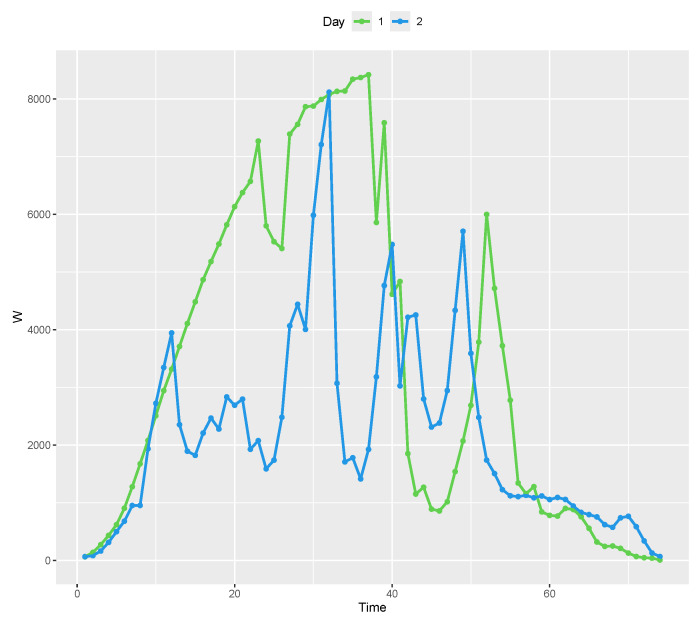
Solar power generated over time for days 1 and 2.

**Figure 3 entropy-26-00824-f003:**
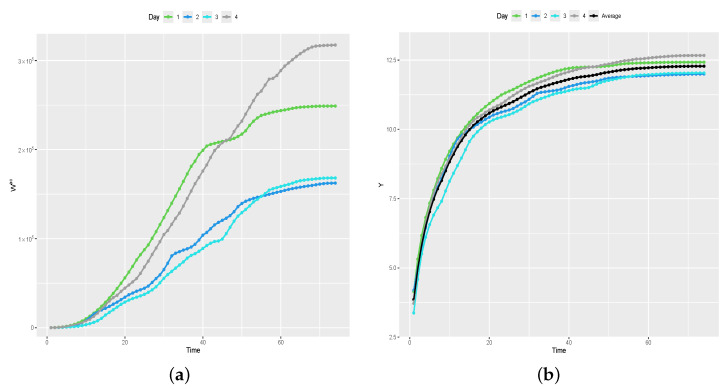
Solar power generated over time. (**a**) $W_{1}$, $W_{2}$, $W_{3}$, and $W_{4}$. (**b**) $W_{1}^{ac}$, $W_{2}^{ac}$, $W_{3}^{ac}$, and $W_{4}^{ac}$.

**Figure 4 entropy-26-00824-f004:**
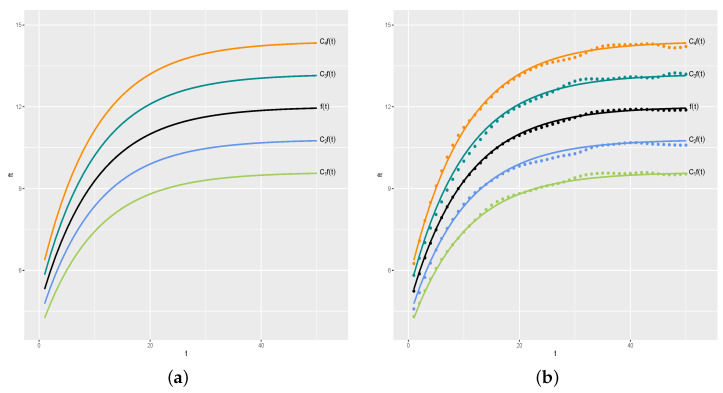
Real curves and generated values. (**a**) Real curves. (**b**) Generated values.

**Figure 5 entropy-26-00824-f005:**
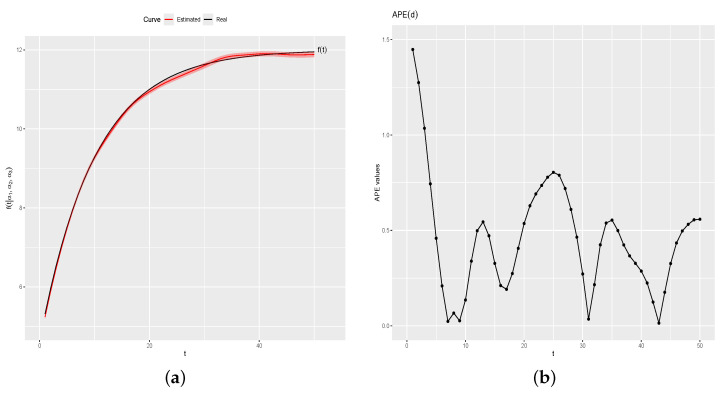
Real and estimated curve and APE(d) values. (**a**) Real and estimated f(t). (**b**) APE(d).

**Figure 6 entropy-26-00824-f006:**
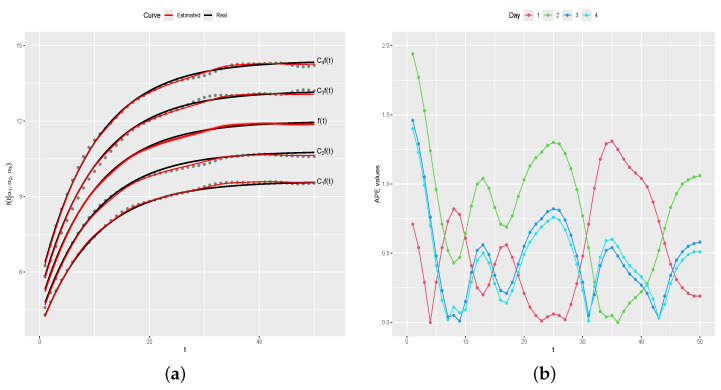
Real and estimated curves and $\mathrm{APE}\left(d_{i}\right)$ values, for i=1,2,3,4. (**a**) Real and estimated curves. (**b**) $\mathrm{APE}\left(d_{i}\right)$ values.

**Figure 7 entropy-26-00824-f007:**
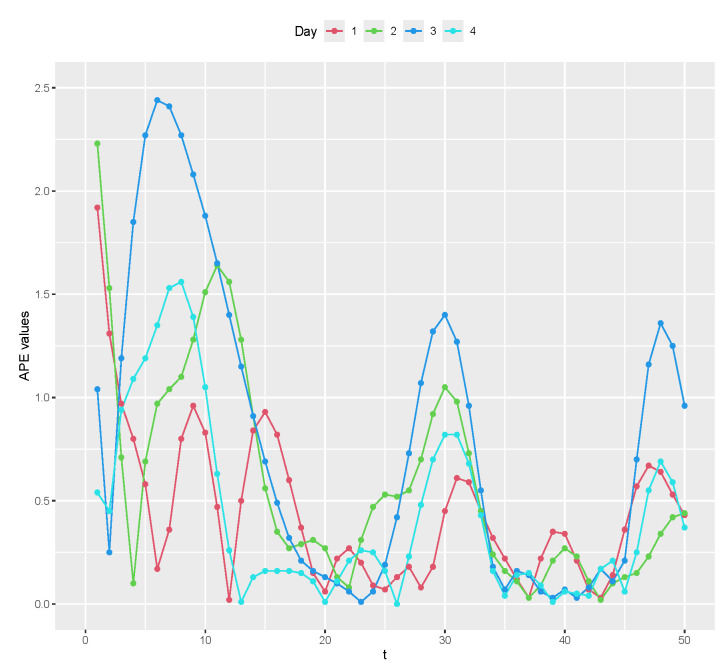
$\mathrm{APE}(e_{i})$ values, for i=1,2,3,4.

**Figure 8 entropy-26-00824-f008:**
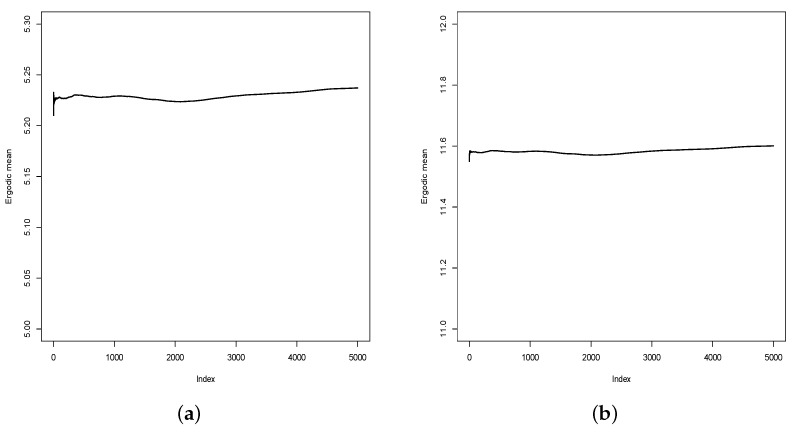
Ergodic mean (ErM) for sampled values for f(1) and f(30). (**a**) f(1). (**b**) f(30).

**Figure 9 entropy-26-00824-f009:**
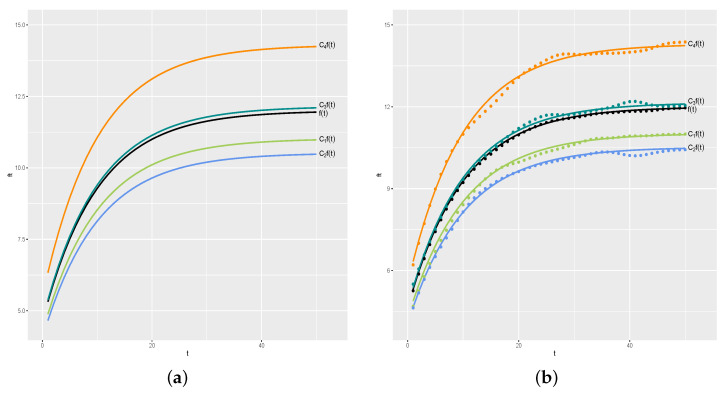
Real curves and generated values. (**a**) Real curves. (**b**) Generated values.

**Figure 10 entropy-26-00824-f010:**
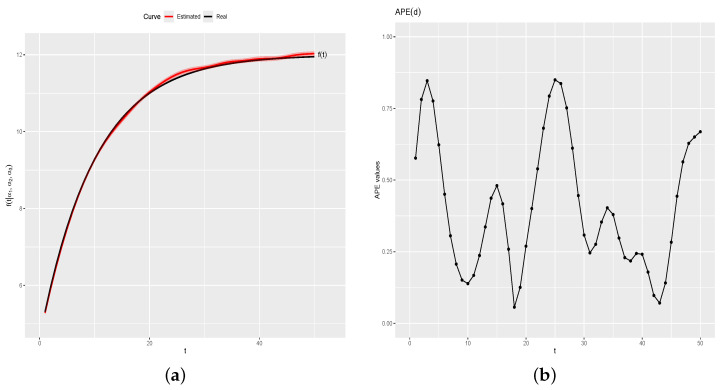
Real and estimated f(t) and APE(d) values. (**a**) Real and estimated curves. (**b**) APE(d) values.

**Figure 11 entropy-26-00824-f011:**
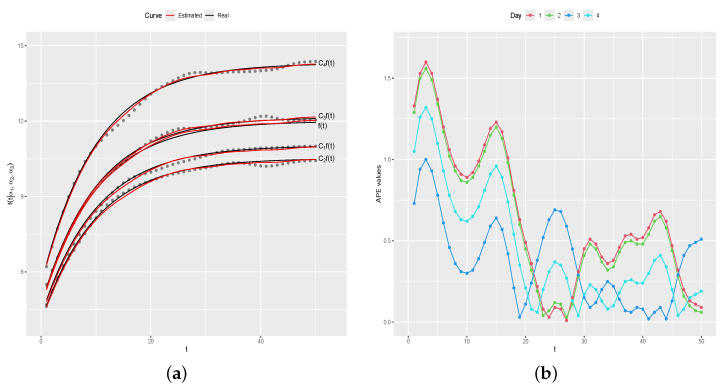
Real and estimated curves and $\mathrm{APE}(d_{i})$ values, for i=1,2,3,4. (**a**) Real and estimated curves. (**b**) $\mathrm{APE}(d_{i})$ values.

**Figure 12 entropy-26-00824-f012:**
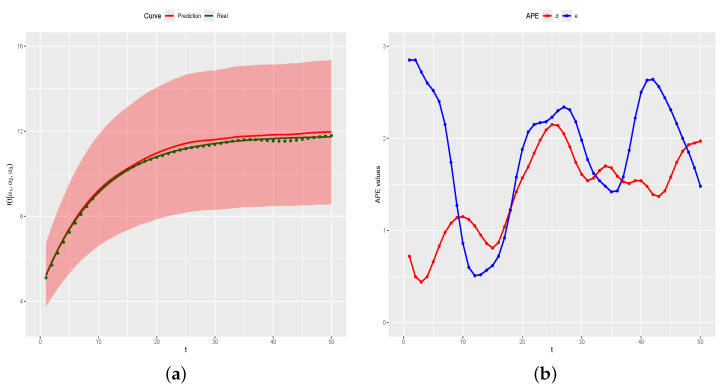
Real and predicted curves and APE values. (**a**) Real and predicted curves. (**b**) APE values.

**Figure 13 entropy-26-00824-f013:**
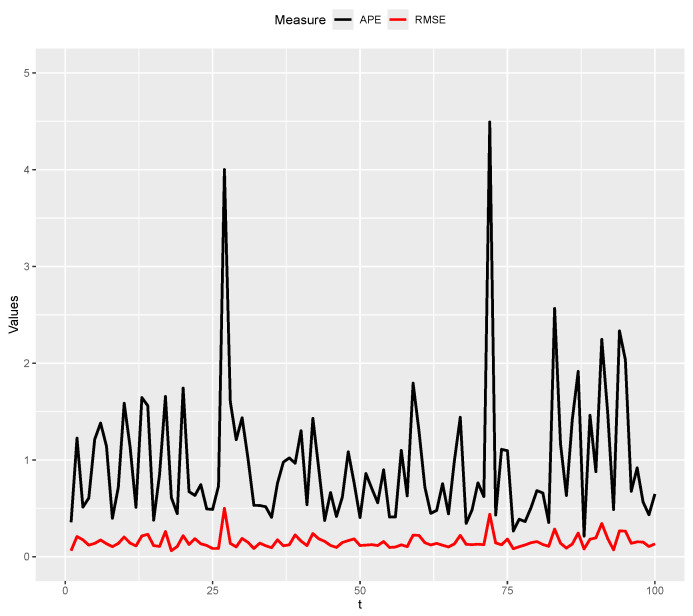
APE and RMSE values.

**Figure 14 entropy-26-00824-f014:**
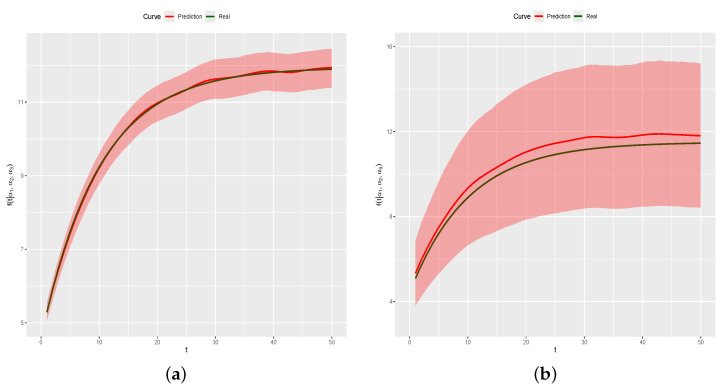
Real and predicted curves for the 18th and 27th simulations. (**a**) 18th. (**b**) 27th.

**Figure 15 entropy-26-00824-f015:**
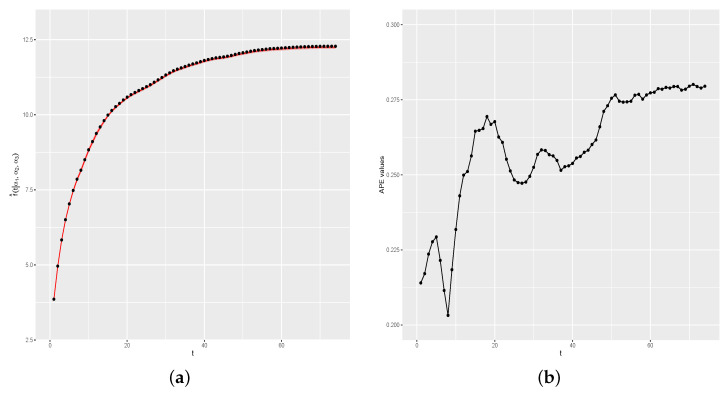
Estimated curve of f(t) and APE(e) values. (**a**) Estimated curve of f(t). (**b**) APE(e) values.

**Figure 16 entropy-26-00824-f016:**
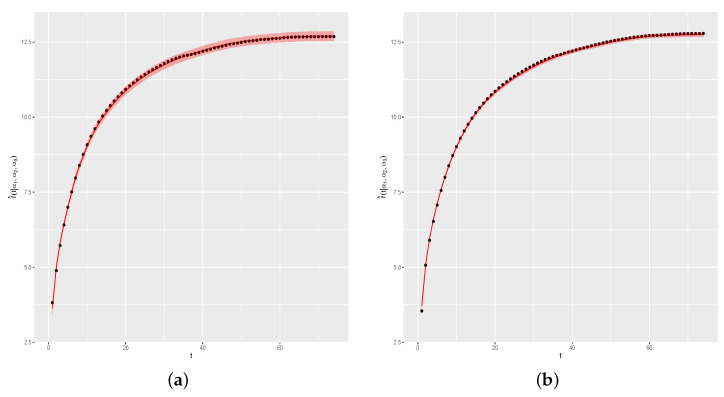
Recorded and predicted values for days 9 and 19. (**a**) Day 9. (**b**) Day 19.

**Figure 17 entropy-26-00824-f017:**
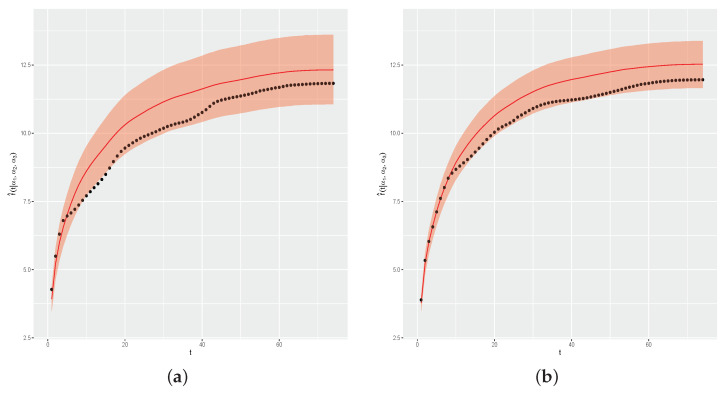
Recorded and predicted values for days 13 and 14. (**a**) Day 13. (**b**) Day 14.

**Table 1 entropy-26-00824-t001:** Real value, estimated value, and 95% credibility interval for $C_{i}$, i=1,2,3,4.

Parameter	Real Value	Estimated Value	Credibility Interval of 95%
$C_{1}$	0.8	0.8060	(0.7997, 0.8080)
$C_{2}$	0.9	0.8955	(0.8913, 0.9003)
$C_{3}$	1.1	1.0998	(1.0939, 1.1057)
$C_{4}$	1.2	1.2006	(1.1949, 1.2071)

**Table 2 entropy-26-00824-t002:** Summary measures of $\mathrm{APE}(d_{i})$ values, for i=1,2,3,4.

Measure	Min	$1^{o}Q$	Median	Mean	$3^{o}Q$	Max
$\mathrm{APE}(d_{1})$	0	0.2100	0.4450	0.5212	0.7675	1.3100
$\mathrm{APE}(d_{2})$	0	0.4825	0.8350	0.7972	1.0575	1.9400
$\mathrm{APE}(d_{3})$	0.0100	0.2400	0.4650	0.4650	0.5775	1.4600
$\mathrm{APE}(d_{4})$	0.0100	0.2375	0.4250	0.4364	1.5750	1.4000

**Table 3 entropy-26-00824-t003:** Summary measures of $\mathrm{APE}(e_{i})$ values, for i=1,2,3,4.

Measure	Min	$1^{o}Q$	Median	Mean	$3^{o}Q$	Max
$\mathrm{APE}(e_{1})$	0.0200	0.1775	0.3600	0.4446	0.6075	1.9200
$\mathrm{APE}(e_{2})$	0.0200	0.2150	0.4300	0.5860	0.9175	2.2300
$\mathrm{APE}(e_{3})$	0.0100	0.1450	0.6200	0.7934	1.2650	2.4400
$\mathrm{APE}(e_{4})$	0	0.1325	0.2400	0.4330	0.3675	1.5600

**Table 4 entropy-26-00824-t004:** MAPE and RMSE values for analyses 1 to 15.

Analysis	MAPE	RMSE	Analysis	MAPE	RMSE	Analysis	MAPE	RMSE
1	0.2590	0.0292	6	0.2038	0.0228	11	0.5360	0.0612
2	0.1395	0.0155	7	0.3670	0.0393	12	0.1815	0.0208
3	0.0500	0.0061	8	0.2378	0.0273	13	0.0095	0.0011
4	0.0753	0.0085	9	0.2157	0.0243	14	0.3445	0.0394
5	0.3996	0.0450	10	0.4903	0.0542	15	0.0146	0.0017

**Table 5 entropy-26-00824-t005:** MAPE and RMSE values for the predictions for days 5 to 19.

Day	MAPE	RMSE	Day	MAPE	RMSE	Day	MAPE	RMSE
5	2.8659	0.3180	10	2.8855	0.3323	15	0.6193	0.07321
6	2.3021	0.2628	11	5.1500	0.5780	16	0.5880	0.0677
7	5.1190	0.5321	12	3.6303	0.4415	17	0.5753	0.0786
8	1.0693	0.1222	13	7.1648	0.7410	18	0.7374	0.1128
9	0.3507	0.0409	14	5.1875	0.6027	19	0.3344	0.0382

## Data Availability

The real dataset is freely available on the websites cited in the article. It also can be obtained by emailing the authors.

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
