# Peer review of "A Bayesian Approach for Modeling and Forecasting Solar Photovoltaic Power Generation"

_entropy, 2024, doi:10.3390/e26100824_

Round 1

Reviewer 1 Report

Comments and Suggestions for Authors

- The authors attempt to use a k-variate normal distribution to model the data. However, based on their explanation, k seems to be 19 as a k represents a day. If this is not correct, the authors need a better explanation.

- After the data were converted to Wi, there seemed not many variations left, only the increasing semi-concave curves were observed. That means it's questionable why a statistical model is needed. Only a mathematical model is sufficient.

- The proposed model should be compared with other models and methods such as VAR and ARIMA.

- In terms of model comparison, the authors should use the marginal likelihood as well.

- In making a prediction, the data need to be split into the training and test sets. In the paper, the APEs seem to be calculated from the in-sample fits, which are not appropriate. Out-of-sample forecast accuracy evaluation is needed.

- To assess the forecast accuracy, the authors should also use RMSE.

Comments on the Quality of English Language

The paper is well-written and organized. However, typos and errors still exist.

Author Response

We would like to thank the reviewer very much for the detailed review of the manuscript and for the comments, suggestions and criticisms in the review report. In the new version of the manuscript, we have modified the article, including following all suggestions and comments made by the three reviewers. Modifications to the text made in response to suggestions from the reviewers are presented in blue in the text.   Please find below point-by-point replies to the comments made by Reviewer 1.  

Reviewer 1:  The authors attempt to use a $k$-variate normal distribution to model the data.  However, based on their explanation, $k$ seems to be 19 as a $k$ represents a day.  If this is not correct, the authors need a better explanation.

Answer: We thank you for this comment. In the new version of the article we have rewritten the part in which we introduce the notation $k$ and $N$ in order to make clear to the readers the difference between the number of measurements per day $k$ and the number of days $N$. Specifically, we inserted the following text: \textcolor{blue}{The dataset used to make inferences on the parameters of the proposed model contains measurements of solar photovoltaic power generation taken at $k=74$ time instants each day over a period of $N=19$ days. In other words, the dataset is a spreadsheet composed of three columns and $k\times n = 74\times 19 = 1,406$ lines. Figure~1 shows a clipping from the data spreadsheet, showing that the first column contains the day (1-19), the second column the time instant (TI, 1-74), and the third column the observed values for photovoltaic solar power (PSP) generated.}   Please see the revised first paragraph of Section~2 of the new version of the article. Since we are modelling the vector of values $\mathbf{Y}_i=\left(Y_{i1},\ldots,Y_{ik}\right)$ of the $i$-th day, which has dimension $k$ , we use a $k$-variate normal distribution for the random errors of the additive model, with $k=74$ and $i=1,\ldots,N=19$.

Reviewer 1: After the data were converted to $W_i$, there seemed not many variations left, only the increasing semi-concave curves were observed. That means it's questionable why a statistical model is needed. Only a mathematical model is sufficient.

Answer: Thank you for this comment. We opt to model the logarithm of the accumulated values due to its measures presenting more stable and predictable behaviour, and also to avoid modelling values on the scale of 100,000 (original scale of the accumulated values) and thus avoid computational problems. Your affirmation that ``\textit{[o]nly a mathematical model is sufficient}'' is very close to the thinking that motivated us to propose our approach, because, as discussed in the article (see the third paragraph of Section 2.1) there may be multiple mathematical functions that can fit the recorded values equally well. Our statistical approach allows us to keep the modelling flexible and avoid the analysis being restricted to a specific chosen mathematical model, giving us a way to estimate the curve of the unknown mathematical function $f$ using the observed data and without imposing a specific functional form.

Reviewer 1: The proposed model should be compared with other models and methods such as VAR and ARIMA. In terms of model comparison, the authors should use the marginal likelihood as well.
Answer: Thank you for this comment. The ARIMA model predicts a given time series based on its own past values. Thus, to apply the ARIMA model to our real dataset, we would need firstly to join the recorded data on the $n$ days into a single time series. However, for the dataset $\mathbf{y}$ used to make inferences, such joining of the recorded values does not lead to a continuous time series, as illustrated in Figure 1 below. Due to this, we believe that tools like ARIMA are not the right ones for our problem. However, we thank the reviewer for asking us this question, because it has led us to think about how to implement a comparison between our model and models already known. We intend to develop this in future work.  

Reviewer 1:  In making a prediction, the data need to be split into the training and test sets. In the paper, the APEs seem to be calculated from the in-sample fits, which are not appropriate. Out-of-sample forecast accuracy evaluation is needed. To assess the forecast accuracy, the authors should also use RMSE.

Answer: As the referee can see in the section Application, we applied the proposed approach for the $N=19$ days by using a ``window'' of $n=4$ days to estimate the curve of $f()$ and predict the curve of the day $(n+1)$. That is, we split the data set from 5 consecutive days into a training dataset (composed by the data of the four first days) and a test dataset (data from the $(n+1)^\text{th}$ day). Using this procedure we have performed fifteen analyses using the $N=19$ days of data. Our first analysis was done using the data from the five first days, separated into the data from the four first days (training dataset) used to fit the model and to predict the curve of the $5^\text{th}$ day (test dataset). The second analysis was done using the datas from day 2 to day 6, separated into the dataset of the days from 2 to 5 (training dataset) used to fit the model and to predict the curve of the $6^\text{th}$ day (test dataset). APE values were calculated by using the recorded values in the sixth day (not used in the fit and prediction procedure) and the predictions made by the procedure using the data from days 2 to 5. We proceed like this until the last analysis, in which we have used the data from days 15 to 19, separated into the data from days 15 to 18 (training dataset) used to fit the model and to predict the curve of the $19^\text{th}$ day (test dataset). APE values were calculated by using the recorded values in the nineteenth day (not used in the fit and prediction procedure) and the predictions made according to our procedure using the data set from days 15 to 18.   We have included the RMSE values for the results obtained in the real dataset.   Please see Tables~4 and 5 of the revised version of the article.  

Reviewer 2 Report

Comments and Suggestions for Authors

Title: A Bayesian approach for modelling and forecasting solar

       photovoltaic power generation

By: Mariana Villela Flesh, Carlos Alberto de Bragança Pereira, Erlandson Ferreira Saraiva

Submitted to: Entropy,   Ms I.d. Entropy-3132342

Report     08/06/2024

The authors propose a Gaussian process prior Bayesian approach for the analysis of the

power generation of the sun in consequetive days. A Gibbs sampling algorithm is used

for the computation of the posterior means. The method is evaluated by simulation

studies.

Major Comments:

* The Gaussian process prior is a well known method for nonparametric Bayesian 

  estimation, and the Gibbs sampling is a common way to compute the posterior mean

  (the Bayes estimator). So it is unclear what's the new part in this paper.

* For the Gaussian process prior approach, it is known that the effect of the prior 

  mean function cannot be eliminated as the data sample increases, thus the posterior

  mean is often an inconsistent estimate of the true mean. This is different from the 

  finite dimension parameter case, in which the posterior mean is always consistent to 

  the true parameter(s). How do you handel this issue?

* The English needs improvement. For example in the Abstract, "the curve of a function"

  better as "the curve" or "the function";  "the historical of" better as  "the history of". 

Comments on the Quality of English Language

see my report

Reviewer 3 Report

Comments and Suggestions for Authors

The paper is an interesting one and, in my opinion, deserves to be published. As expected, I have some comments listed below.

I agree with you, parametric forms are rather limited and I appreciate your idea of including a GP as a prior making things non-linear. However, you have still embedded the GP inside a parametric framework (Gaussian and Wishart distributions). Henceforth I would ask you to be a bit more loose on your terminology and mention something along the lines of a semi-parametric approach, because after all it is such. 

Secondly, in your simulation studies you should predict the future and not estimate how accurately you fit the curve. So, fit the model in some data and then predict future values and compute the MAPE  (and why not MAE as well?) in those future predicted values. I see you did this in the real data, so you must do it in the simulation studies as well, otherwise the results of the simulation studies are misleading.

Comments on the Quality of English Language

The authors should proof-read the paper, as there are typos and grammatical errors. With no intention to offend, when I read, in the second page, the “An usual…” I understood that you must be Spanish. A couple of lines later, the paragraph finishes with “…still remains.” While it should be “…still remain.” since the adjective is plural (“issues still remain”, not “remains”). There are a lot of these types of mistakes and lessen the image of the paper, which I have to say, in general is very well written and well presented.

Round 2

Reviewer 1 Report

Comments and Suggestions for Authors

The authors made a considerable attempt to revise the manuscript and it's publishable now. I have no further comments.

Comments on the Quality of English Language

N/A

Reviewer 2 Report

Comments and Suggestions for Authors

I read the response letter, and still think the contribution is not enough for publication in Entropy.

Reviewer 3 Report

Comments and Suggestions for Authors

I see the authors addressed my comments. I am happy with the new version.

Comments on the Quality of English Language

Please have a proof-read of the paper. I noticed some minor typos throughout the manuscript.